# Recombinant protein KR95 as an alternative for serological diagnosis of human visceral leishmaniasis in the Americas

Mahyumi Fujimori[1], Ruth Tamara Valencia-Portillo[1], José Angelo Lauletta Lindoso[2,3], Beatriz Julieta Celeste[1,4], Roque Pacheco de Almeida[5], Carlos Henrique Nery Costa[6], Alda Maria da Cruz[7], Angelita Fernandes Druzian[8], Malcolm Scott Duthie[9], Carlos Magno Castelo Branco Fortaleza[10], Ana Lúcia Lyrio de Oliveira[8], Anamaria Mello Miranda Paniago[8], Igor Thiago Queiroz[11], Steve Reed[9], Aarthy C. Vallur[12], Hiro Goto[1,4‡], Maria Carmen Arroyo Sanchez[1,4‡]*

1 Instituto de Medicina Tropical, Faculdade de Medicina, Universidade de São Paulo, São Paulo, São Paulo, Brazil, 2 Departamento de Doenças Infecciosas e Parasitárias, Faculdade de Medicina, Universidade de São Paulo, São Paulo, São Paulo, Brazil, 3 Instituto de Infectologia Emílio Ribas, Secretaria de Estado da Saúde, São Paulo, São Paulo, Brazil, 4 Departamento de Medicina Preventiva, Faculdade de Medicina, Universidade de São Paulo, São Paulo, São Paulo, Brazil, 5 Departamento de Medicina Interna e Patologia, Hospital Universitário/EBSERH, Universidade Federal de Sergipe, Aracaju, Sergipe, Brazil, 6 Instituto Natan Portella para Doenças Tropicais, Universidade Federal do Piauí, Teresina, Piauí, Brazil, 7 Laboratório Interdisciplinar de Pesquisas Médicas, Instituto Oswaldo Cruz/FIOCRUZ, Rio de Janeiro, Rio de Janeiro, Brazil, 8 Faculdade de Medicina, Universidade Federal de Mato Grosso do Sul, Campo Grande, Mato Grosso do Sul, Brazil, 9 HDT Bio, Seattle, Washington, United States of America, 10 Departamento de Doenças Tropicais e Diagnóstico por Imagem, Universidade Estadual Paulista Júlio de Mesquita Filho, Botucatu, São Paulo, Brazil, 11 Hospital Giselda Trigueiro, Secretaria Estadual da Saúde Pública, Natal, Rio Grande do Norte, Brazil, 12 InBios International Inc, Seattle, Washington, United States of America

☯ These authors contributed equally to this work.
‡ HG and MCAS also contributed equally to this work.
* arroyo@usp.br

**Data Availability Statement:** All relevant data are within the manuscript and its Supporting information files.

## Abstract

In the Americas, visceral leishmaniasis (VL) is caused by the protozoan *Leishmania infantum*, leading to death if not promptly diagnosed and treated. In Brazil, the disease reaches all regions, and in 2020, 1,933 VL cases were reported with 9.5% lethality. Thus, an accurate diagnosis is essential to provide the appropriate treatment. Serological VL diagnosis is based mainly on immunochromatographic tests, but their performance may vary by location, and evaluation of diagnostic alternatives is necessary. In this study, we aimed to evaluate the performance of ELISA with the scantily studied recombinant antigens, K18 and KR95, comparing their performance with the already known rK28 and rK39. Sera from parasitologically confirmed symptomatic VL patients (n = 90) and healthy endemic controls (n = 90) were submitted to ELISA with rK18 and rKR95. Sensitivity (95% CI) was, respectively, 83.3% (74.2–89.7) and 95.6% (88.8–98.6), and specificity (95% CI) was 93.3% (85.9–97.2) and 97.8% (91.8–99.9). For validation of ELISA with the recombinant antigens, we included samples from 122 VL patients and 83 healthy controls collected in three regions in Brazil (Northeast, Southeast, and Midwest). When comparing the results obtained with the VL patients' samples, significantly lower sensitivity was obtained by rK18-ELISA (88.5%, 95% CI: 81.5–93.2) compared with rK28-ELISA (95.9%, 95% CI: 90.5–98.5), but the sensitivity

**Funding:** This study was supported by Laboratório de Investigação Médica (LIM 38) Hospital das Clínicas da Faculdade de Medicina da Universidade de Sao Paulo (https://limhc.fm.usp.br/portal/). HG received a research fellowship (n°: 302940/2019-7) from Conselho Nacional de Desenvolvimento Científico e Tecnológico – CNPq (https://www.gov.br/cnpq/pt-br). MF received a scholarship (n°: 2017/03367-3) from Fundação de Amparo à Pesquisa do Estado de São Paulo (https://fapesp.br/). RTVP received a scholarship (n°: 88882.376665/2019-01 and n°: 88887.689454/2022-00) from Coordenação de Aperfeiçoamento de Pessoal de Nível Superior (https://www.gov.br/capes/pt-br). The funders had no role in study design, data collection, analysis, publication decision, or manuscript preparation.

**Competing interests:** The authors have declared that no competing interests exist.

was similar comparing rKR95-ELISA (95.1%, 95% CI: 89.5–98.0), rK28-ELISA (95.9%, 95% CI: 90.5–98.5), and rK39-ELISA (94.3%, 95% CI: 88.4–97.4). Analyzing the specificity, it was lowest with rK18-ELISA (62.7%, 95% CI: 51.9–72.3) with 83 healthy control samples. Conversely, higher and similar specificity was obtained by rKR95-ELISA (96.4%, 95% CI: 89.5–99.2), rK28-ELISA (95.2%, 95% CI: 87.9–98.5), and rK39-ELISA (95.2%, 95% CI: 87.9–98.5). There was no difference in sensitivity and specificity across localities. Cross-reactivity assessment, performed with sera of patients diagnosed with inflammatory disorders and other infectious diseases, was 34.2% with rK18-ELISA and 3.1% with rKR95-ELISA. Based on these data, we suggest using recombinant antigen KR95 in serological assays for VL diagnosis.

## Introduction

Leishmaniases are diseases of public health relevance, considered among the most neglected diseases globally, with more than 350 million people living in areas at risk. These diseases are caused by the obligate intracellular protozoan parasite of the *Leishmania* genus and are transmitted through the infected female phlebotomine sandflies [1, 2]. They present in different clinical forms, but visceral leishmaniasis (VL) is the most severe and potentially fatal when untreated, potentially leading to death in 95% of cases [3]. In 2020, World Health Organization reported 12,838 VL cases worldwide (12,739 autochthonous and 99 imported); 16% were concentrated in the Americas [4]. In Brazil, VL is caused by *Leishmania infantum*. In 2020, Brazil reported 1,933 new VL cases, representing more than 97% of cases in the Americas, and the lethality for the same period was 9.5% [5, 6]. Since the clinical presentation of VL is not pathognomonic, diagnostic tests must be performed to confirm the diagnosis, and the patients must be treated immediately; thus, accurate methods are essential.

Parasitological techniques are the gold standard for laboratory diagnosis of VL, but although highly specific, their sensitivity is dependent on the organ source [7]. For example, in a bone marrow aspirate sample, the sensitivity reaches 52–85% [8], while with spleen aspirates, the sensitivity varies from 93–99% [8, 9]; however, the latter is not authorized in Brazil for ethical reasons. Thus, searching for a less invasive and easy-to-perform method with high sensitivity and specificity has leveraged diagnostic research. In addition, amid technological advances, studies on recombinant proteins in detecting anti-*Leishmania* antibodies have been increasingly used as an alternative in diagnosing VL [10–14].

The antigen rK39 contains a 39-amino acid repeat that is part of the LcKin protein predominant in *L. chagasi* amastigotes and closely related to *L. donovani* [15]. The immunochromatographic test with the recombinant protein K39 represents an advance in the diagnosis of VL due to its low cost, combined with fast and easy execution, with high sensitivity (100%) and specificity (98.0%) [16]. However, in a study evaluating rapid test performance in different regions, sensitivity reached 75.5% and specificity 70.0% in Ethiopia. A similar result was found in the meta-analysis performed by Chappuis [17], in which lower sensitivity was observed in East Africa than in South Asia. These results corroborate those that Sanchez [18] found, evaluating different regions in Brazil. Due to the low sensitivity of the rK39 antigen in Eastern Africa, an *L. donovani* fusion polyprotein was synthesized and named rK28 [19]. Although high sensitivities (94–98%) and specificities (95–98%) were obtained with rK28 in studies in Sudan and Ethiopia, the number of samples evaluated was relatively small [20].

Due to the difference in the accuracy of the immunochromatographic tests using the recombinant antigens observed among the various regions where VL occurs, studies evaluating alternative antigens are necessary, aiming for efficient tests for VL diagnosis, and new antigens are being developed. Thus, in the present study, the performance of two recombinant antigens not incorporated in VL diagnosis, rK18, and rKR95, was evaluated in an ELISA format and compared with the known recombinant proteins rK28 and rK39. The validation of rKR95-ELISA showed high sensitivity and specificity and similar performance compared with rK28 and rK39 in samples from different regions of Brazil. Thus, the rKR95 antigen is a good alternative for diagnosing human VL.

## Materials and methods

### Study design

The performance of two recombinants, rK18 and rKR95, was evaluated in an ELISA format and compared to the known recombinant proteins rK28 and rK39 at the Instituto de Medicina Tropical, Faculdade de Medicina, Universidade de São Paulo (USP) (IMT/FMUSP) between 2021 and 2022. We employed stored samples from healthy asymptomatic individuals, VL patients, VL/AIDS co-infected patients, and patients with other diseases that may cross-react with VL antigens. Except for the other disease samples, all were collected in different Brazilian leishmaniasis endemic areas from the Northeast, Midwest, and Southeast.

### Ethics statement

The Research Ethics Committee of Hospital das Clinicas da Faculdade de Medicina da Universidade de São Paulo (USP) approved the project (number 5.602.042). Since the 639 samples employed in this study were obtained from the Biorepository belonging to the Instituto de Medicina Tropical, Faculdade de Medicina, USP, and under the responsibility of Hiro Goto and Maria Carmen Arroyo Sanchez, the need for participant consent was waived by the ethics committee. The samples were coded to preserve the individual's anonymity. Furthermore, the research results were disclosed so that the individual could not be correlated with the said result. We also guarantee the ethically correct use of the material and the information obtained from it.

### Study subjects

The 639 samples employed in this study were distributed in three panels.

Panel 1—Patients and healthy controls. They consisted of 90 samples from confirmed VL cases and 90 samples of healthy endemic controls with follow-up. The *Leishmania* parasite was detected in VL patients in bone marrow aspirates [21, 22]. All samples, except one, were positive by direct agglutination test (DAT), with variable titers: high, medium, or low [23]. The patients were from Campo Grande/MS (n = 42) and Piaui State (n = 48) (S1 Fig; S1 and S2 Tables), Midwest and Northeast, respectively. In addition, healthy subjects from an area with *Leishmania* transmission had negative DAT and remained asymptomatic for VL for six months. These individuals were from Tres Lagoas/MS, Midwest (S1 Fig; S1 and S2 Tables). With this panel (panel 1), we calculated sensitivity, specificity, and cut-off.

Panel 2—Patients and healthy controls from different geographical regions in Brazil. These patients were from the Northeast, Midwest, and Southeast regions. Two hundred sixty-nine samples were divided into three groups. a) 122 samples from VL patients whose *Leishmania* infection was confirmed by microscopic examination of the bone marrow aspirate or by positive DAT. The active leishmaniasis was confirmed at the collection site before they were

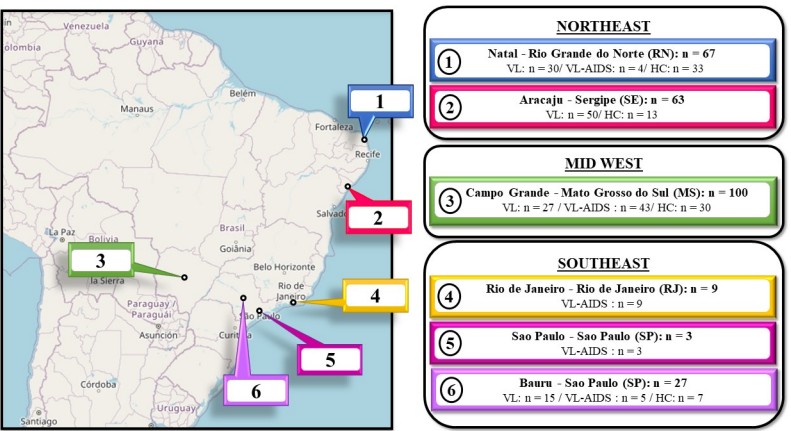

**Fig 1. Patients and healthy controls from different geographical regions in Brazil.** Samples were collected from 122 VL patients, 64 VL-AIDS patients, and 83 healthy individuals living in endemic areas. n—number of samples; VL—visceral leishmaniasis; VL-AIDS—co-infection; HC—healthy controls. The base map was obtained from https://apps.nationalmap.gov/viewer/.

included in the study, and all patients presented at least two of the following symptoms or signs of VL: fever, hepatomegaly, splenomegaly, and cytopenia. These were from Natal/RN (city 1) (n = 30), Aracaju/SE (city 2) (n = 50), Campo Grande/MS (city 3) (n = 27), and Bauru/SP (city 6) (n = 15) (Fig 1 and S2 Fig; S3 and S4 Tables). b) 64 samples from VL/AIDS co-infected patients were characterized using routine diagnostic methods in each center. These samples were from Natal/RN (city 1) (n = 4), Campo Grande/MS (city 3) (n = 43), Rio de Janeiro/RJ (city 4) (n = 9), Sao Paulo/SP (city 5) (n = 3), and Bauru/SP (city 6) (n = 5) (Fig 1 and S2 Fig; S3 and S4 Tables). c) 83 samples from healthy individuals living in endemic areas and DAT negative. These individuals were from Natal/RN (city 1) (n = 33), Aracaju/SE (city 2) (n = 13), Campo Grande/MS (city 3) (n = 30), and Bauru/SP (city 6) (n = 7) (Fig 1 and S2 Fig; S3 and S4 Tables).

Panel 3—potentially cross-reactive controls. They consisted of 190 samples from patients with other diseases confirmed for autoimmune disease (n = 10), Chagas disease (n = 47), cutaneous leishmaniasis (n = 28), malaria (n = 12), mucosal leishmaniasis (n = 14), paracoccidioidomycosis (n = 27), syphilis (n = 20), toxoplasmosis (n = 20), active pulmonary tuberculosis (n = 12) (S3 Fig and S5 Table).

With samples from panels 2 and 3, we validated the ELISA test using the recombinant proteins K18 and KR95.

The samples were coded and analyzed randomly to prevent the risk of bias.

## Sample size calculation

Aiming at a 95% sensitivity for the rK18-ELISA and rKR95-ELISA tests [24] and considering a 95% confidence interval (95% CI) ± 5%, the minimum number required for this study was 73 VL patients. Aiming at a 98% specificity and considering a 95% CI of ± 3%, the minimum number required was 84 healthy controls [24, 25].

## Serological tests

Direct Agglutination Test (DAT) based on *L. donovani* promastigotes was produced by *KIT Biomedical Research*—Amsterdam, The Netherlands. IT-Leish based on rK39 antigen was

produced by *DiaMed AG*—Cressier-sur-Morat, Switzerland, and *Bio-Rad Laboratories*—Marnes-la-Coquette, France. Kalazar-Detect based on rK39 antigen was produced by *InBios International*—Seattle, WA, USA. All tests were performed at Instituto de Medicina Tropical, Faculdade de Medicina, USP, by the time the samples were collected, according to the manufacturer's recommendations.

For DAT, samples were two-fold diluted from 1:100 (final 1:200) through 1:102,400 (final 1:204,800), and a cut-off of 1:3,200 was adopted [18, 23, 26, 27]. All healthy endemic control samples were DAT negative (titer < 1:3,200). However, VL patients with negative DAT were included in the study as long as they were positive in the parasitological test.

## Recombinant antigens

The recombinant antigens, K28, K39, K18, and KR95, were purchased from Infectious Disease Research Institute (IDRI), Seattle, United States. To better evaluate the recombinant antigens K18 and KR95, the already known rK28 and rK39 proteins were employed.

rK39 (*L. infantum*—syn. *chagasi*) is part of a large protein kinesin-related (Lc-Kin), containing 298 amino acids and has a molecular mass of 38.9 kD [15].

rK28 (*L. donovani*) is a fusion polyprotein comprising HASPB1 (*L. infantum* K26 homolog), LdK39 (*L. infantum* K39 homolog), and HASPB2 (*L. infantum* K9 homolog) and has a molecular mass of 28.33 [19].

rKR95 (*L. donovani*) is a kinesin-related protein with a molecular mass of 95 kD, presenting 100% identity with *L. infantum* [28].

rK18 (*L. infantum*—syn. *chagasi*) is a tandem repeat hypothetical protein (also known as rTR18) with a molecular mass of 18 kD, presenting 100% identity with *L. donovani* [29].

## Enzyme-linked immunosorbent assay—ELISA

ELISA standardization of the four recombinants was performed to maintain the same laboratory conditions and make a more reliable comparison. The standardization of the human ELISA protocol was based on the standardization performed by Fujimori [30] for canine samples.

In summary, high-binding polystyrene plates of 96 wells (entire area) (Corning 3590, Incorporated, New York, USA) were coated separately with 50 μL/well of each antigen. Dilutions of 0.5 μg/mL for rK39 and 1 μg/mL for rK28, rKR95, and rK18 in 0.06M carbonate/bicarbonate buffer pH 9.6 were used. The plates were incubated at 4ºC in a humid chamber overnight and washed three times with 0.01M phosphate-buffered saline (PBS), pH 7.2 containing 0.05% Tween 20 (Polyoxyethylene sorbitan monolaurate, Sigma-Aldrich, St. Louis, USA) (PBS-T). The plates were blocked with PBS-T containing 5% skimmed milk (Molico, Nestlé) (PBS-T-L 5%) (125 μL/well) at 37ºC for 2 hours in a humid chamber, followed by three washes with PBS-T. For standardization, two positive (parasitologically positive) and one negative (DAT negative) serum sample was diluted 1/25, 1/50, and 1/100 in PBS-T containing 2% and 5% skimmed milk (PBS-T-L 2% and 5%). The sample dilutions in duplicate (50 μL/well) were applied, and the plates were incubated at 37˚C in a humid chamber for 30 minutes, followed by five washes with PBS-T. Then, 50 μL/well of goat anti-human IgG peroxidase conjugate (Sigma A-0170, Saint Louis, MO, USA) diluted 1/10,000, 1/20,000, 1/30,000, and 1/40,000 in PBS-T-L 2% was added and incubated at 37˚C in a moist chamber for 30 min, followed by five washes with PBS-T. The development of the reaction was done with TMB/$H_2O_2$ (100 μL/well) (Tetramethylbenzidine/hydrogen peroxide) chromogen (Novex-Life Technologies, Carlsbad, CA, USA) (50 μL/well) at different times: 5, 7, 10, and 15 minutes, at room temperature. The development of the reaction was stopped by adding 25 μL/well $H_2SO_4$ 2N. The absorbances

were read at 450 nm using an ELISA reader (Multiskan Go-Thermo Scientific, Finland). Throughout the study, samples were assayed in duplicate.

### Sensitivity, specificity, and cut-off calculation

Each recombinant antigen was tested in ELISA against 180 samples (90 VL patients and 90 healthy endemic controls) from sera panel 1 (S1 Fig). The absorbance values obtained by each assay with specific recombinant antigens were transformed into the *Percentage of absorbance of the positive standard* (*ABS% Positive*) by dividing the absorbance of each sample by the absorbance of the positive standard serum and multiplying by 100 [31]. ROC (receiver operating characteristic) curve was constructed for each recombinant antigen from the *ABS% Positive* values calculated for the VL-confirmed patients and healthy control samples from the endemic area. The cut-off of each recombinant was set for optimal sensitivity, specificity, and accuracy. Each sample's reactivity index (RI) was calculated as the quotient between the ABS% Positive and the cut-off. RI $\geq$ 1 was considered positive.

### Validation of rK18-ELISA and rKR95-ELISA with samples from VL patients and healthy controls

For the validation step, additional samples were included to evaluate each recombinant protein. As a result, the sensitivity, specificity, and accuracy values were calculated for 122 samples from VL patients, 64 samples from VL/AIDS co-infected patients, 83 samples from healthy individuals living in endemic areas and negative by DAT, and 190 samples from patients with other diseases.

### Statistical analysis

The results were analyzed using the software R version 4.2.1 for Windows, RStudio version 2022.02.3 for Windows, GraphPad Prism, version 9.3.1 for Windows (GraphPad Software Inc., San Diego, CA, USA), GraphPad/ Quickcalcs (https://www.graphpad.com/quickcalcs/confInterval1/) and MedCalc (© 2022 MedCalc Software Ltd, https://www.medcalc.org/calc/).

To test for the normal distribution of the values, we used the Shapiro-Wilk test, and to test the homogeneity of variances, we used the Levene test. In addition, continuous variables were compared by the Mann-Whitney test (two independent groups), Friedman test (three or more matched groups), and Kruskal-Wallis's test (three or more independent groups). Finally, Dunn's multiple comparisons test was made when statistically significant differences were observed.

Sensitivity, specificity, accuracy, positive and negative likelihood ratios, and diagnostic odds ratios with 95% confidence intervals (95% CI) were calculated. Cochran's Q test compared the sensitivity and specificity among the recombinant antigens. When statistically significant differences were observed, multiple comparisons were made with pairwise McNemar's chi-square test, and adjusted p-values were considered. Pearson's chi-square test compared proportions. The agreement of the results of the recombinant antigens (two by two) was performed using the *Kappa* index, calculated according to Fleiss [32], and interpreted according to Landis and Koch [33]: *Kappa* < 0: No agreement, 0.00–0,20: Slight agreement; 0.21–0.40: Fair agreement; 0.41–0.60: Moderate agreement; 0.61–0.80: Substantial agreement; 0.81–1.00: Almost perfect agreement. Finally, the results of the recombinant antigens (two by two) were compared using Spearman's correlation test.

A significance level of 0.05 was considered (p < 0.05).

## Results

### Evaluation of the performance of recombinant antigen ELISAs for the diagnosis of human visceral leishmaniasis

Analyzing the results of the various ELISA test conditions with recombinant antigens, we defined the best concentration of antigen as 0.5 μg/mL (rK39) and 1.0 μg/mL (rK18, rK28, and rKR95); 1/100 serum dilution for the four antigens; conjugate dilution 1/10,000 for the four antigens; chromogen time of 7 minutes (rK28, rK39, and rKR95) and 10 minutes (rK18). After defining the conditions for best performance, the cut-off point was calculated based on ROC curves constructed using the values of the *ABS% Positive Control* of each sample (Fig 2). The optimal cut-off point was 10 with rK28-ELISA, 7 with rK39-ELISA, 1.155 with rK18-ELISA, and 7 with rKR95-ELISA. The four antigens discriminated significantly between negative (healthy controls) and positive (VL patients) samples (Mann-Whitney test: p < 0.0001). Nevertheless, the cut-off obtained by rK18-ELISA showed that the absorbance values were much lower than those with the other antigens.

We observed significant differences in the *ABS% Positive* in sera from VL patients and healthy controls using the different recombinant antigens (Friedman test: p < 0.0001). In VL patients, the rK28-ELISA, rK39-ELISA, and rKR95-ELISA gave significantly higher median *ABS% Positive* values (Dunn's multiple comparisons test: p < 0.0001) than rK18-ELISA (Fig 3). In healthy controls, compared with rK18-ELISA, all antigens gave higher median *ABS% Positive* values (Dunn's multiple comparisons test: p < 0.0001 compared with rK28-ELISA, and rKR95-ELISA; p = 0.0013 compared with rK39-ELISA). Also, no significant difference was observed in rKR95-ELISA results compared with rK39-ELISA (Dunn's: p = 0.0517) and rK28 (Dunn's: p > 0.9999) in control samples (Fig 3).

The heat map correlation between the pairs of antigens evaluated in the ELISA test using samples from VL patients and healthy controls showed a very strong Spearman correlation (ρ) between positive and negative results obtained by rK39-ELISA and rK28-ELISA (ρ = 0.94; 95% CI: 0.93–0.96), rK39-ELISA and rKR95-ELISA (ρ = 0.93; 95% CI: 0.91–0.95), and rK28-ELISA

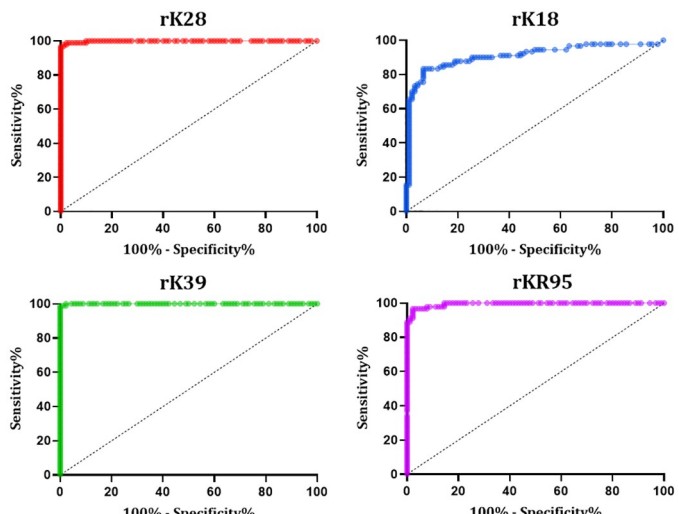

**Fig 2. Receiver operating characteristic (ROC) curves to determine ELISA performance.** ROC curves were constructed using the values of *ABS% Positive* obtained from testing samples of VL patients (n = 90) and healthy endemic controls (n = 90) (Panel 1).

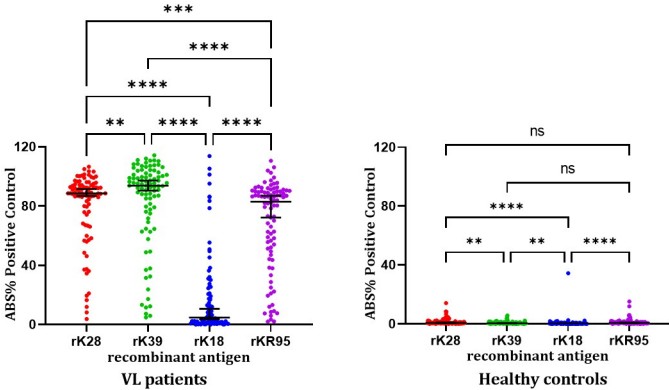

**Fig 3. Percentage of absorbance of the positive standard (*ABS% Positive*) obtained by ELISA using recombinant antigens in sera from VL patients and healthy controls (Panel 1).** The horizontal black lines represent the median ABS% Positive Control values with a 95% confidence interval. Friedman test: p < 0.0001 (VL patients and Healthy controls). Dunn's multiple comparisons test: VL patients: **—p = 0.0049; ***—p = 0.0002; ****—p < 0.0001. Healthy controls: **—p = 0.0044 (rK28 x rK39) and p = 0.0013 (rK18 x rK39); ****—< 0.0001; ns—p = 0.0517 (rK39 x rKR95) and p > 0.9999 (rK28 x rKR95).

and rKR95-ELISA (ρ = 0.92; 95% CI: 090–0.94), indicating that the rKR95 antigen has similar diagnostic performance compared with rK39 and rK28 (S4 Fig).

Table 1 presents the diagnostic performance obtained by rK28-ELISA, rK39-ELISA, rK18-ELISA, and rKR95-ELISA. Considering the 90 VL patients (panel 1), a significant difference was obtained in sensitivity (Cochran's Q test: p < 0.0001). Lower values were obtained by rK18-ELISA (83.3%; 95% CI: 74.2%-89.7%) compared with rK28-ELISA (97.8%; 95% CI: 91.8%-99.9%) (pairwise McNemar's test: p = 0.0019), rK39-ELISA (97.8%; 95% CI: 91.8%-99.9%) (pairwise McNemar's test: p = 0.0024), and rKR95-ELISA (95.6%; 95% CI: 88.8%-98.6%) (pairwise McNemar's test: p = 0.0046). There was no significant difference in

**Table 1. Diagnostic performance parameters obtained by ELISA with recombinant antigens determined by ROC curves (Panel 1).**

| Antigens | Number of individuals | | | | Diagnostic performance (95% CI) | | | | | |
|---|---|---|---|---|---|---|---|---|---|---|
| | TP | FN | TN | FP | Sensitivity % [a] | Specificity % [b] | LR + | LR - | Accuracy % [a] | Diagnostic Odds Ratio |
| **rK28** | 88 | 2 | 89 | 1 | 97.8 (91.8–99.9) | 98.9 (93.4–100.0) | 88.00 (12.53–618.11) | 0.02 (0.01–0.09) | 98.3 (95.0–99.7) | 3916.0 (348.7–43973.5) |
| **rK39** | 88 | 2 | 90 | 0 | 97.8 (91.8–99.9) | 100.0 (95.6–100.0) | NC | 0.02 (0.01–0.09) | 98.9 (95.8–100.0) | 6407.4 (303.3–135365.4) |
| **rK18** | 75 | 15 | 84 | 6 | 83.3 [c] (74.2–89.7) | 93.3 (85.9–97.2) | 12.50 (5.74–27.23) | 0.18 (0.11–0.28) | 88.3 [d] (82.8–92.3) | 70.0 (25.8–189.6) |
| **rKR95** | 86 | 4 | 88 | 2 | 95.6 (88.8–98.6) | 97.8 (91.8–99.9) | 43.00 (10.91–169.42) | 0.05 (0.02–0.12) | 96.7 (92.8–98.6) | 946.0 (168.9–5300.1) |

ROC (receiver operating characteristic) curves were constructed using the values of *ABS% Positive* obtained from testing samples of VL patients (n = 90) and healthy endemic controls (n = 90).

95% CI—95% probability confidence interval; TP—true positive; FN—false negative; TN—true negative; FP—false positive; LR+—Positive likelihood ratio; LR-—Negative likelihood ratio; NC—not calculated.

Cochran's Q test:

[a]—p < 0.0001;

[b]—p = 0.0189

Pairwise McNemar's test:

[c]—p = 0.0019—compared with rK28-ELISA; p = 0.0024—compared with rK39-ELISA; p = 0.0046—compared with rKR95-ELISA.

[d]—p = 0.0001—compared with rK28-ELISA and rK39-ELISA; p = 0.0021—compared with rKR95-ELISA.

the sensitivity of rKR95 with the recombinant proteins, rK28 (pairwise McNemar's test: p = 0.3800) and rK39 (pairwise McNemar's test: p = 0.1530) in the ELISA test.

Although a significant difference was obtained in specificity in the group of 90 healthy controls (panel 1) (Cochran's Q test: p = 0.0189), rK18-ELISA specificity (93.3%; 95% CI: 85.9%-92.2%) was not significantly lower compared with rK39-ELISA (100.0%; 95% CI: 95.1%-100.0%) (pairwise McNemar's test: p = 0.0759), rK28-ELISA (98.9%; 95% CI: 93.4–100.0%) (pairwise McNemar's test: p = 0.0759), rK95-ELISA (97.8%; 95% CI: 91.8–99.9%) (pairwise McNemar's test: p = 0.2360).

The agreement of the diagnostic performance (*kappa*—κ) was *substantial* between rK18-ELISA and rK28-ELISA (κ = 0.800; 95% CI = 0.712–0.887), rK18-ELISA and rK39-ELISA (κ = 0.766; 95% CI = 0.672–0.860), rK18-ELISA and rKR95-ELISA (κ = 0.766; 95% CI = 0.672–0.860). *Almost perfect* agreement was obtained between rK28-ELISA and rK39-ELISA (κ = 0.944; 95% CI = 0.896–0.992), rK28-ELISA and rKR95-ELISA (κ = 0.922; 95% CI = 0.866–0.976) and rK39-ELISA and rKR95-ELISA (κ = 0.933; 95% CI = 0.881–0.986).

Considering VL patients and healthy controls (panel 1) (Table 1), a significant difference was obtained in accuracy among antigens (Cochran's Q test: p < 0.0001). The highest accuracy (95% CI) was achieved with rK28-ELISA: 98.3% (95.0–99.7) (pairwise McNemar's test: p = 0.0001), rK39-ELISA: 98.9% (95.8–100.0) (pairwise McNemar's test: p = 0.0001) and rKR95-ELISA: 96.7% (92.8–98.6) (pairwise McNemar's test: p = 0.0021), compared with rK18-ELISA: 88.3% (82.8–92.3). Similar accuracy was achieved with rK28-ELISA (pairwise McNemar's test: p = 0.3080) and rKR95-ELISA (pairwise McNemar's test: p = 0.1530) compared with rK39-ELISA. These results show that the rKR95 antigen has high accuracy and can be an alternative for VL diagnosis. Further, in this preliminary evaluation, the performance of the rK18-ELISA was lower than the other ELISA tests.

## Validation of recombinant antigen ELISA for the diagnosis of human visceral leishmaniasis

In validating recombinant antigens ELISA, additional sera were used to search for *Leishmania* infection, including samples from different geographical localities in Brazil. Also, the interference of antibodies against other diseases that may cross-react in serological tests for VL was investigated in the sera of patients diagnosed with inflammatory disorders and other infectious diseases.

We compared the reactivity indices obtained by ELISA with the four recombinant antigens testing 122 samples from VL patients, 64 samples from VL / AIDS patients, 83 samples from healthy controls, and 190 samples from patients with inflammatory disorders and other infectious diseases. In each group of samples, a significantly different distribution of reactivity indices was obtained with the antigens (Friedman test: p < 0.0001, followed by Dunn's multiple comparisons test) (Fig 4).

In VL patients, higher median RI was obtained by rK39-ELISA compared with rK18-ELISA (Dunn's: p < 0.0001), rK28-ELISA (Dunn's: p < 0.0001), and rKR95-ELISA (Dunn's: p = 0.0282). Also, higher values were obtained by rKR95-ELISA compared with rK28-ELISA (Dunn's: p = 0.0004).

In VL / AIDS patients, lower median RI was observed in rK18-ELISA compared with rK39-ELISA (Dunn's: p < 0.0001) and rKR95-ELISA (Dunn's: p = 0.0300); higher median values were obtained by rK39-ELISA compared with rKR28-ELISA (Dunn's: p > 0.0077); no differences was observed between rKR95-ELISA and rKR39-ELISA (Dunn's: p = 0.2825) and rK28-ELISA (Dunn's: p > 0,9999).

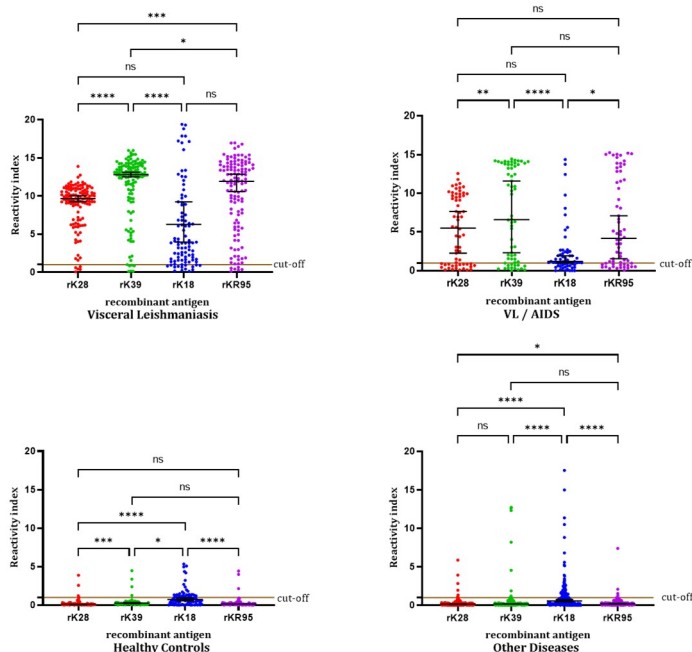

**Fig 4. Reactivity indices obtained by ELISA with recombinant antigens with 122 samples from VL patients, 64 samples from VL / AIDS patients, 83 samples from healthy controls (Panel 2), and 190 samples from patients with other diseases (Panel 3).** The horizontal black lines represent the median *ABS% Positive Control* values with a 95% confidence interval. Friedman test: $p < 0.0001$ (for the four groups of samples). Dunn's multiple comparisons test: VL patients: *—$p = 0.0282$; ***—$p = 0.0004$; ****—$p < 0.0001$; ns—$p = 0.2522$ (rK18 x rKR95) and $p = 0.3187$ (rK28 x rK18). VL / AIDS patients: *—$p = 0.0300$; **—$p = 0.0077$; ****—$p < 0.0001$; ns—$p = 0.6920$ (rK28 x rK18), $p > 0.9999$ (rk28 x rKR95) and $p = 0,2825$ (rK39 x rKR95). Healthy controls: *—$p = 0.0374$; ***—$p = 0.0003$; ****—$p < 0.0001$; ns —$p = 0.4002$ (rK28 x rKR95) and $p = 0.1692$ (rK39 x rKR95). Other diseases patients: *—$p = 0.0152$; ****—$p < 0.0001$; ns—$p = 0.1646$ (rK28 x rK39) and $p > 0.9999$ (rK39 x rKR95). The four antigens discriminated significantly between negative (healthy controls) and positive (VL patients) samples (Mann-Whitney test: $p < 0.0001$).

In healthy control individuals, a higher median RI was obtained by rK18-ELISA compared with rK28-ELISA (Dunn's: $p < 0.0001$), rK39-ELISA (Dunn's: $p = 0.0374$), and rKR95-ELISA (Dunn's: $p < 0.0001$); lower RI were seen in rK28-ELISA compared with rK39-ELISA (Dunn's: $p = 0.0003$); no difference was obtained between rKR95-ELISA and rK28-ELISA (Dunn's: $p = 0.4002$) and rKR95-ELISA and rK39-ELISA (Dunn's: $p = 0.1692$).

In the group of patients with inflammatory disorders and other infectious diseases, a higher median RI was obtained by rK18-ELISA compared with rK28-ELISA (Dunn's: $p < 0.0001$), rK39-ELISA (Dunn's: $p < 0.0001$), and rKR95-ELISA (Dunn's: $p < 0.0001$); lower values were seen in rKR95-ELISA compared with rK28-ELISA (Dunn's: $p = 0.0152$); no difference was obtained between rK28-ELISA and rK39-ELISA (Dunn's: $p = 0.1646$) and rKR95-ELISA and rK39-ELISA (Dunn's: $p > 0.9999$).

We also compared the RI obtained by ELISA in the groups of samples tested with each recombinant antigen and observed a significantly different distribution of values (Kruskal-Wallis's: $p < 0.0001$, followed by Dunn's multiple comparisons test) (Fig 5).

With rK28-ELISA, a higher median RI was obtained in the VL patients group compared with healthy control individuals (Dunn's: $p < 0.0001$), VL / AIDS patients (Dunn's: $p = 0.0327$), and patients with other diseases (Dunn's: $p < 0.0001$); higher median RI were obtained in VL / AIDS patients compared with healthy control individuals (Dunn's: $p < 0.0001$) and other diseases (Dunn's: $p < 0.0001$); no difference was obtained between healthy control individuals and other diseases (Dunn's: $p = 0.6551$).

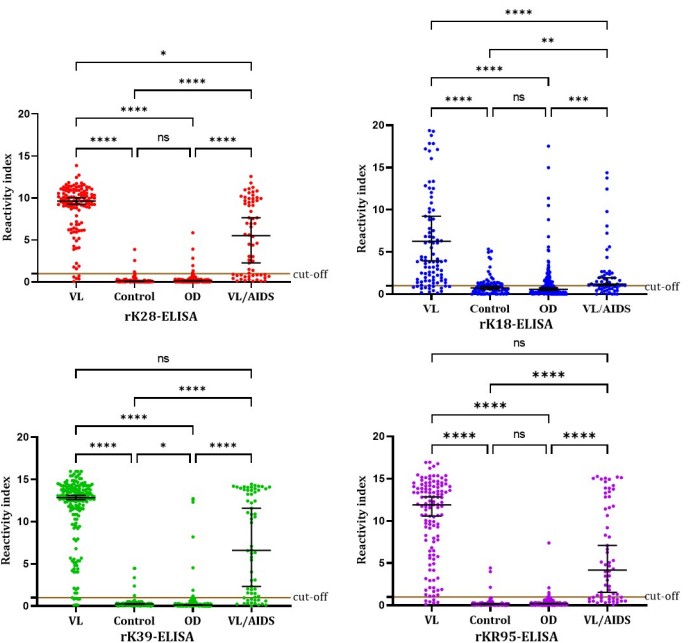

**Fig 5. Reactivity indices obtained by rK28-ELISA, rK39-ELISA, rK18-ELISA, and rKR95-ELISA with 122 samples from VL patients, 83 samples from healthy controls, 190 samples from patients with other diseases, and 64 samples from VL / AIDS patients.** The horizontal black lines represent the median *ABS% Positive Control* values with a 95% confidence interval. Kruskal-Wallis's: p < 0.0001 (for the four antigens). Dunn's multiple comparisons test: rK28: *—p = 0.0327; ****—p < 0.0001; ns—p = 0.6551. rK39: *—p = 0.0262; ****—p < 0.0001; ns—p = 0.0515. rK18: **—p = 0.0025; ***—p = 0.0003; ****—p < 0.0001; ns—p > 0.9999. rKR95: ****—p < 0.0001; ns—p > 0.9999 (control x other diseases) and p = 0.1357 (VL x VL / AIDS).

With rK39-ELISA, a higher median RI was obtained in the VL patients group compared with healthy control individuals (Dunn's: p < 0.0001) and patients with other diseases (Dunn's: p < 0.0001); a higher median RI was obtained in VL / AIDS patients compared with healthy control individuals Dunn's: (p < 0.0001) and other diseases (Dunn's: p < 0.0001); a higher RI was obtained with healthy control individuals compared with other diseases (Dunn's: p = 0.0262); no difference was obtained between VL and VL / AIDS patients (Dunn's: p = 0.0515).

With rK18-ELISA, a higher median RI was obtained in the VL patients group compared with healthy control individuals (Dunn's: p < 0.0001), VL / AIDS patients (Dunn's: p < 0.0001), and patients with other diseases (Dunn's: p < 0.0001); a higher median RI was obtained in VL / AIDS patients compared with healthy control individuals (Dunn's: p = 0.0025) and other diseases (Dunn's: p = 0.0003); no difference was obtained between healthy control individuals and other diseases (Dunn's: p > 0.999).

With rkR95-ELISA, a higher median RI was obtained in the VL patients group compared with healthy control individuals (Dunn's: p < 0.0001) and patients with other diseases (Dunn's: p < 0.0001); a higher median RI was obtained in VL / AIDS patients compared with healthy control individuals (Dunn's: p < 0.0001) and other diseases (Dunn's: p < 0.0001); no difference was obtained between healthy control individuals and other diseases (Dunn's: p > 0.999) and between VL and VL / AIDS patients (Dunn's: p = 0.1357).

These comparisons showed that rKR95 and rK39 were the only antigens that did not show significant differences between VL and VL / AIDS patients.

Table 2 presents the performance of the recombinant antigens. Comparing the performance obtained by rK28-ELISA, rK39-ELISA, rK18-ELISA, and rKR95-ELISA in the 122 samples from VL patients (panel 2), a significant difference in sensitivity (Cochran's Q test: p = 0.0018) was obtained. Sensitivity with rK18-ELISA (88.5%; 95% CI: 81.5%-93.2%) was significantly lower compared with rK28-ELISA (95.9%; 95% CI: 90.5–98.5%) (pairwise McNemar's test: p = 0.0400). No difference was observed between rK18-ELISA and rK39-ELISA (94.3%; 95% CI: 88.4–97.4%) (pairwise McNemar's test: p = 0.0696), and rKR95 (95.1%; 95% CI: 89.5–98.0%) (pairwise McNemar's test: p = 0.0627). Sensitivity with rKR95-ELISA was similar to rK28-ELISA (pairwise McNemar's test: p = 0.5640) and rK39-ELISA (pairwise McNemar's test: p = 0.3800). On the other hand, no significant difference was observed with rK28-ELISA, rK39-ELISA, rK18-ELISA, and rKR95-ELISA in the 64 samples from VL / AIDS patients (Cochran's Q test: p = 0.1378).

In the 83 samples from healthy controls (panel 2), a significant difference in specificity was observed (Cochran's Q test: p < 0.0001). The lowest specificity was obtained by rK18-ELISA (62.7%; 95% CI: 51.9%-72.3%) compared with rK28-ELISA (95.2%; 95% CI: 87.9%-98.5%) (pairwise McNemar's test: p < 0.0001), rK39-ELISA (95.2%; 95% CI: 87.9%-98.5%) (pairwise McNemar's test: p < 0.0001) and rKR95-ELISA (96.4%; 95% CI: 89.5%-99.2%) (pairwise McNemar's test: p < 0.0001). In turn, specificity with rKR95-ELISA (96.4%) was similar compared with rK28-ELISA (95.2%) and rK39-ELISA (95.2%) (pairwise McNemar's test: p = 0.6770).

In the group of 190 samples from patients with other diseases (panel 3), a significant difference in specificity was observed (Cochran's Q test: p < 0.0001). The lowest specificity was obtained by rK18-ELISA (65.8%; 95% CI: 58.8%-72.2%) compared with rK28-ELISA (95.3%; 95% CI: 91.1%-97.6%) (pairwise McNemar's test: p < 0.0001), rK39-ELISA (95.8%; 95% CI: 91.8%-98.0%) (pairwise McNemar's test: p < 0.0001), and rKR95-ELISA (96.8%; 95% CI: 93.1%-98.7%) (pairwise McNemar's test: p < 0.0001). On the other hand, specificity with rKR95-ELISA (96.8%) was similar compared with rK28-ELISA (95.3%) (pairwise McNemar's

**Table 2. Performance obtained by rK18-ELISA, rK28-ELISA, rK39-ELISA, and rKR95-ELISA in sera from VL patients, VL / AIDS patients, healthy controls (Panel 2), and patients with other diseases (Panel 3).**

| Antigens | Sensitivity% (n)—95% CI | | | | Specificity% (n)—95% CI | | | | Diagnostic Odds Ratio—95% CI | |
|---|---|---|---|---|---|---|---|---|---|---|
| | VL patients [a] | | VL /AIDS patients [b] | | Healthy controls [c] | | Other diseases [c] | | VL patients x Health controls | |
| rK28 | 95.9 (117) | 90.5–98.5 | 68.7 (44) | 56.6–78.8 | 95.2 (79) | 87.9–98.5 | 95.3 (181) | 91.1–97.6 | 462 | 95–2,245 |
| rK39 | 94.3 (115) | 88.4–97.4 | 76.6 (49) | 64.8–85.3 | 95.2 (79) | 87.9–98.5 | 95.8 (182) | 91.8–98.0 | 324 | 73–1,428 |
| rK18 | 88.5 (108) [d] | 81.5–93.2 | 64.1 (41) | 51.8–74.7 | 62.7 (52) [e] | 51.9–72.3 | 65.8 (125) [e] | 58.8–72.1 | 12 | 5–29 |
| rKR95 | 95.1 (116) | 89.5–98.0 | 71.9 (46) | 59.8–81.5 | 96.4 (80) | 89.5–99.2 | 96.8 (184) | 93.1–98.7 | 515 | 97–2,717 |

VL patients—n = 122, VL/AIDS patients—n = 64, Health controls—n = 83, Other diseases—n = 190.

95% CI—95% probability confidence interval; n—number of positive samples (VL and VL/AIDS patients) / number of negative samples (healthy controls and other diseases).

Cochran's Q test:

[a]—p = 0.0018;

[b]—p = 0.1378;

[c]—p < 0.0001.

Pairwise McNemar's test:

[d]—p = 0.0400—compared with rK28-ELISA.

[e]—p < 0.0001—compared with rK28-ELISA, rK39-ELISA, and rKR95-ELISA.

test: p = 0.3800) and rK39-ELISA (95.8%) (pairwise McNemar's test: p = 0.3800). No difference was observed in the specificity between rK28-ELISA and rK39-ELISA (pairwise McNemar's test: p = 0.7050).

Considering VL patients and healthy controls, the agreement of the diagnostic performance (*kappa—*κ) was *moderate* between rK18-ELISA and rK28-ELISA (κ = 0.583; 95% CI = 0.470–0.696), rK18-ELISA and rK39-ELISA (κ = 0.586; 95% CI = 0.474–0.698), rK18-ELISA and rKR95-ELISA (κ = 0.565; 95% CI = 0.451–0.679). *Almost perfect* agreement was obtained between rK28-ELISA and rK39-ELISA (κ = 0.960; 95% CI = 0.921–0.999), rK28-ELISA and rKR95-ELISA (κ = 0.940: 95% CI = 0.892–0.987) and rK39-ELISA and rKR95-ELISA (κ = 0.959; 95% CI = 0.920–0.999).

The heat map correlation between the pairs of antigens evaluated in the ELISA test using samples from VL patients, VL / AIDS patients, healthy controls, and patients with other infectious diseases is in S5 Fig. Very strong Spearman correlation (ρ) was observed between rK39-ELISA and rKR95-ELISA (ρ = 0.92; 95% CI: 0.89–0.94), with VL patients. In addition, a strong Spearman correlation (ρ) was observed between rK28-ELISA and rK39-ELISA (ρ = 0.84; 95% CI: 0.77–0.88) and between rK28-ELISA and rKR95-ELISA (ρ = 0.72; 95% CI: 0.62–0.80) with VL patients; between rK28-ELISA and rK39-ELISA (ρ = 0.74; 95% CI: 0.62–0.82) with VL/AIDS patients; between rK39-ELISA and rKR95-ELISA (ρ = 0.72; 95% IC: 0.57–0.82) and between rK28-ELISA and rKR95-ELISA (ρ = 0.70; 95% CI: 0.55–0.81) with healthy controls; between rK39-ELISA and rKR95-ELISA (ρ = 0.71; 95% CI: 0.63–0.78) with other diseases patients.

Comparing the sensitivity obtained by rK28-ELISA, rK39-ELISA, and rKR95-ELISA in samples from patients with VL from different regions in Brazil (S6 Fig), in city 3, a significant difference was obtained by tests (Cochran's Q test: p = 0.0074). Lower sensitivity was obtained by rK18-ELISA (81.5%; 95% CI: 62.8%-92.3%) compared with rK28-ELISA, rK39-ELISA, and rKR95-ELISA (96.3%; 95% CI: 80.2%-100.0%) (pairwise McNemar's test: p = 0.0455). In cities 1, 2, and 6, no significant difference was observed with rK18-ELISA, rK28-ELISA, rK39-E-LISA, and rKR95-ELISA (Cochran's Q test: p > 0.05). Across localities, no difference was observed in the sensitivity with any recombinant antigen (chi-square test, p > 0.05).

No significant difference in sensitivity (Cochran's Q test; p = 0.1378) was observed with rK28-ELISA, rK39-ELISA, rK18-ELISA, and rKR95-ELISA in samples of patients co-infected with VL and AIDS (S7 Fig) from city 6 (Cochran's Q test: p = 0.3916), city 3 (Cochran's Q test: p = 0.6096), city 1 (Cochran's Q test: p = 0.3916) and city 4 (Cochran's Q test: p = 0.8358). Between localities, no difference was observed in the sensitivity with any recombinant antigen (chi-square test, p>0.05).

Comparing the specificity obtained by rK28-ELISA, rK39-ELISA, rK18-ELISA, and rKR95-ELISA in samples from healthy controls from different regions in Brazil (S8 Fig), a significant difference was obtained by tests in city 3 (Cochran's Q test: p < 0.0001) and city 1 (Cochran's Q test: p < 0.0001). In city 3, a significantly lower specificity (pairwise McNemar's test: p = 0.0002) was obtained by rK18-ELISA (53.3%; 95% CI: 36.1%-69.8%) compared with rK28-ELISA, rK39-ELISA, and rKR95-ELISA (100.0%; 95% CI: 86.5%-100.0%). In city 1, a significantly lower specificity was also obtained by rK18-ELISA (57.6%; 95% CI: 40.8%-72.8%) compared with rK28-ELISA (pairwise McNemar's test: p = 0.0016), rK39-ELISA (pairwise McNemar's test: p = 0.0016) and rKR95-ELISA (pairwise McNemar's test: p = 0.0027) (93.9%; 95% CI: 79.4%-99.3%). No significant difference was observed in cities 2 and 6 with rK18-E-LISA, rK28-ELISA, rK39-ELISA, and rKR95-ELISA (Cochran's Q test: p > 0.05). In addition, no difference across localities (chi-square test, p>0.05) was observed in specificity with any recombinant antigen.

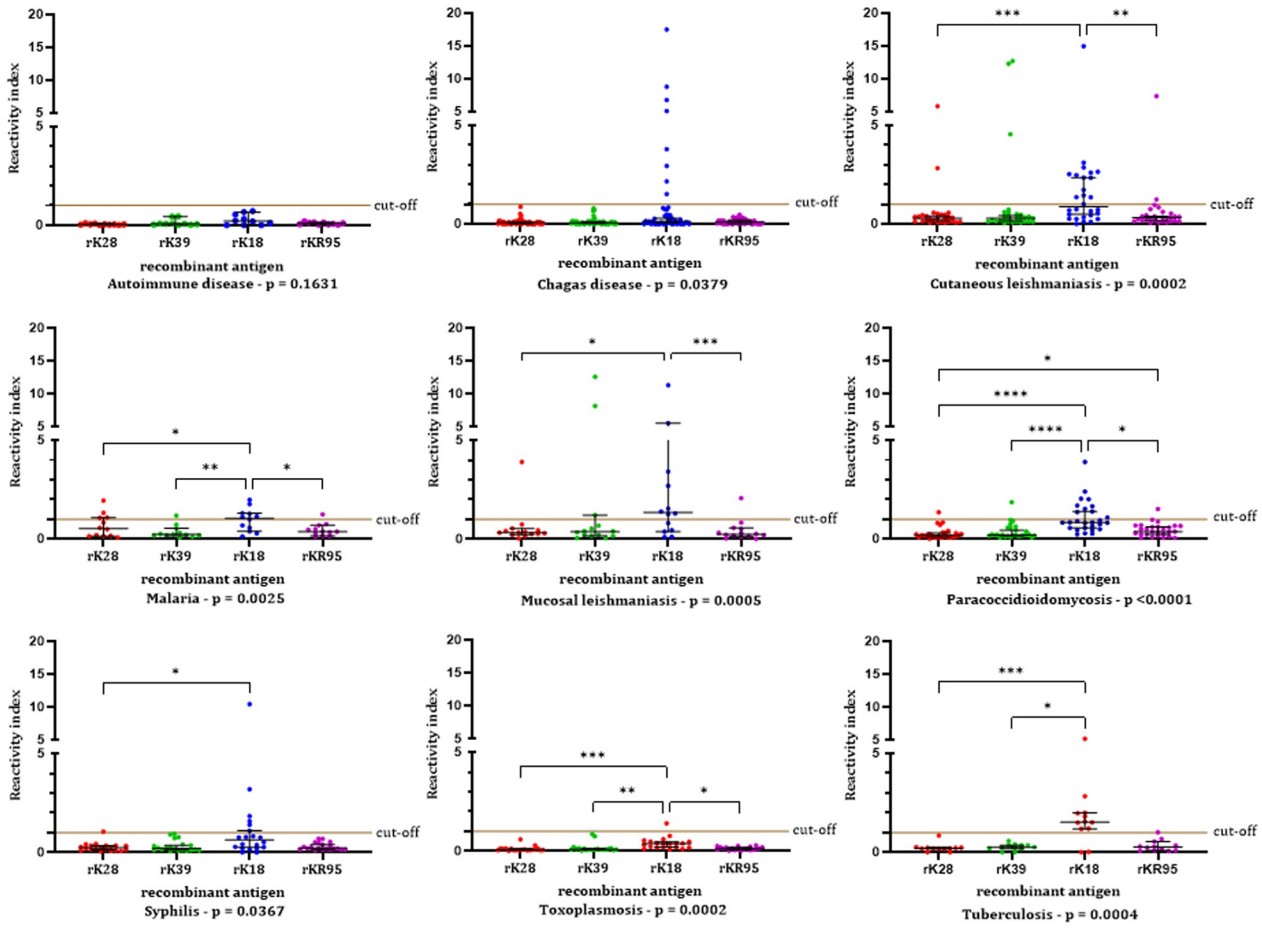

**Fig 6. Antigen-specific antibodies in sera patients with potentially confounding infectious diseases (Panel 3) developed for recombinant antigens.** The horizontal black lines represent the median ABS% Positive Control values with a 95% confidence interval. Friedman test (p values are in the graphic for each disease), followed by Dunn's multiple comparisons test: Cutaneous leishmaniasis: **—p = 0.0038; ***—p = 0.0002. Malaria: *—p = 0.0342 (rK28 x rK18) and p = 0.0170 (rK18 x rKR95); **—p = 0.0011. Mucosal leishmaniasis: *—p = 0.0325; ***—p = 0.0002. Paracoccidioidomycosis: *—p = 0.0225 (r18 x rKR95) and p = 0.0368 (rK28 x rKR95); ****—p < 0.0001. Syphilis: *—p = 0.0291. Toxoplasmosis: *—p = 0.0351; **—p = 0.0057; ***—p = 0.0001. Tuberculosis: *—p = 0.0207; ***—p = 0.0003.

Concerning the results obtained by rK28-ELISA, rK39-ELISA, rK18-ELISA, and rKR95-E-LISA in the sera of patients with other diseases, analyzing each disease separately, Fig 6 presents the RI and S6 Table, the specificity obtained for each potentially confounding infectious disease studied.

Significantly different distribution of reactivity indices (Friedman test) was obtained with the antigens employed in samples of Chagas disease (p = 0.0379), cutaneous leishmaniasis (p = 0.0002), malaria (p = 0.025), mucosal leishmaniasis (p = 0.0005), paracoccidioidomycosis (p < 0.0001), syphilis (p = 0.0367), toxoplasmosis (p = 0.0002), and tuberculosis (p = 0.0004) (Fig 6).

Except for samples from patients with autoimmune disease and toxoplasmosis, rK18-ELISA showed lower specificity than the other antigens. In malaria patients, rK28-ELISA showed low specificity, not statistically different from rK18-ELISA.

## Discussion

In the present study, we aimed to evaluate the performance of two scantily studied recombinant proteins, K18 and KR95, in the serological diagnosis of VL. In our previous study analyzing VL dog samples, rKR95 showed good performance in serological diagnosis [30]. In this study, using samples from a well-characterized panel, we achieved excellent results with the recombinant KR95 protein, with 95.6% sensitivity and 97.8% specificity, close to the results with the already-known rK28 and rK39 proteins. Furthermore, compared with the routine tests used in diagnosing human VL in Brazil, the sensitivity and specificity obtained in the present study were better [34]. While in the study carried out by Freire [34], the tests on the ELISA platform, *Leishmania* ELISA IgG+IgM, Ridascreen *Leishmania* Ab, and NovaLisa *Leishmania infantum* IgG presented a respective accuracy of 85.5%, 85.5%, and 91.2%; the rKR95-ELISA showed an accuracy of 96.7%.

ELISA-rK18 presented the lowest accuracy (88.3%) and diagnostic odds ratio (70.0), with a significant difference in sensitivity compared with rK28-ELISA, rK39-ELISA, and rKR95-E-LISA, and specificity when compared with rK39-ELISA. Nevertheless, these results are close to those of Pedras et al. [27], who obtained diagnostic efficiency of 90.7% in the ELISA test with rK39 protein and 85% in the ELISA test with *Leishmania chagasi* antigen. In addition, a study conducted on *L. donovani*-infected patients from Ethiopia and Bangladesh that employed rK18 in an ELISA format observed that antibodies anti-rK18 declined in the treatment follow-up over 180 days in Ethiopia and 365 days in Bangladesh [35]. However, the authors did not discriminate the sensitivity.

As the performance of serological methods may vary depending on the host factors, HIV co-infection, and the geographic region where the infection was acquired [36–38], samples from VL patients, healthy controls, VL/AIDS co-infected patients, and patients with other diseases from several locations in the country were used to validate the sensitivity and specificity of the ELISA tests.

Comparing the sensitivity of the rKR95-ELISA (95.6%) in the group of VL samples, it was not significantly different from rK39-ELISA (94.3%) and rK28-ELISA (95.9%). Fujimori et al. [30] obtained 95.9% sensitivity in the rKR95-ELISA to diagnose canine leishmaniasis, similar to our results in human infection. In the same study, plates sensitized with recombinant KR95 were evaluated for 180 days in storage at 4°C. There was no significant decrease in reactivity, indicating that this protein is stable for diagnostic kits [30]. Searching for healthy individuals living in endemic areas, the rKR95-ELISA showed the highest specificity (96.4%).

Previous studies showed that variations in sensitivity and specificity might occur when an antigen is tested with samples from different localities, as shown by Sanchez et al. [18], using rapid tests based on rK39. A similar result was observed by Figueiredo et al. [38]. In addition, a global study observed lower sensitivity in East Africa and Brazil compared with the Indian sub-continent region [23]. In the present study, there was no significant difference in the sensitivity and specificity data of rKR95-ELISA comparing samples from the various localities in the country. Therefore, contrary to the results with rK39 in another study by Sanchez et al. [18], the rKR95-based assay would be acceptable in sensitivity to use in different Brazilian regions.

In Bangladesh, rK39, rTR18, and rKR95 antigens were evaluated in an ELISA format for the detection of asymptomatic *L. donovani* infections, and the combination of the results of the three antigens increased the sensitivity (87%) but decreased the specificity (83%) [28].

In the group of other diseases, the specificity of the rKR95-ELISA (96.8%) was higher than with the other antigens. This test did not cross-react with samples from patients with autoimmune disease, Chagas disease, syphilis, and toxoplasmosis. Only one sample from malaria, paracoccidioidomycosis, and tuberculosis was above the cut-off value, indicating that the

rKR95-ELISA may discriminate against several widely distributed diseases in Brazil. Pedras et al. [27] conducted a study that showed 85.0% specificity in the rK39-ELISA for malaria, 100% for syphilis, and 83.3% for Chagas disease. Although the recombinant KR95 protein has a 79% identity with *Trypanosoma cruzi* [28], no cross-reactivity was found in our evaluation.

Concerning the performance of rK28-ELISA, despite the small number of malaria samples, the 66.7% specificity obtained corroborated the results obtained in Ethiopia, where rK28 RDT gave 32.7%-39.6% specificity due to the cross-reactivity with malaria and other diseases [39]. In addition, another study of the same group detected that 18.4% of confirmed malaria patients cross-reacted with rK28 RDT [40]. Therefore, according to the authors, the cross-reactivity between rK28 and anti-*P. falciparum* antibodies may be due to five HASPB1 repeats in *P. falciparum* and rK28 [40].

In the group of samples from patients co-infected with VL/AIDS, although the sensitivity by rKR95-ELISA (71.9%) was higher than by rK28-ELISA (68.7%) and lower than by rK39-ELISA (76.6%), these differences were not significant. Being a group with a low humoral response and antibody production, the sensitivity of the serological test tends to be lower than that of non-HIV co-infected patients [41, 42]. Therefore, the present results are similar to those in the literature [36, 41, 42]. However, rKR95 and rK39 were the only antigens that showed no significant differences in the results with samples from VL and VL / AIDS patients; thus, we may consider the rKR95-ELISA as an attractive alternative for use in VL diagnosis. Furthermore, for use in the point of care, this rKR95 antigen may be considered for developing a lateral flow immunochromatography format.

One limitation of this study was the small number of samples of each disease that are part of panel 3 of cross-reactivity, thus making a more accurate analysis of the recombinant antigens difficult. Another limitation was the impossibility of calculating the predictive value because the samples were selected for convenience, that is, with the previous diagnosis. Another limitation was the heterogeneity of the groups of samples in panel 2, with a smaller number of healthy controls in Aracaju and Bauru and the characterization of VL/AIDS co-infected patients that were different according to the locality of collection.

In summary, we validated the rKR95-ELISA, suggesting the rKR95 antigen as a good alternative for diagnosing human VL.

## Supporting information

**S1 Table. Demographic and laboratory data for VL patients and healthy controls from endemic areas used to construct the ROC curves (Panel 1).** n—number of samples; VL—visceral leishmaniasis; $^{\alpha}$—missing information of one VL patient; M—man; F—female; β—missing information of 16 VL patients; DAT—direct agglutination test; min-max—minimum-maximum; *—band test intensity; $^{\mu}$—21 samples from VL patients and one healthy control sample were not tested in the Kalazar Detect and IT-Leish tests.
(DOCX)

**S2 Table. Collection date and diagnosis criteria for VL patients and healthy controls from endemic areas used to construct the ROC curves.** n—number of samples; VL—visceral leishmaniasis; DAT—direct agglutination test.
(DOCX)

**S3 Table. Demographic and laboratory data for VL patients, VL / AIDS patients, and healthy controls from various Brazilian endemic areas (Panel 2).** n—number of samples; VL—visceral leishmaniasis; VL/AIDS—co-infection; $^{\alpha}$—missing information of three VL patients and 41 VL/AIDS patients; M—man; F—female; $^{\beta}$—missing information of five VL

patients and 41 VL/AIDS patients; min-max—minimum-maximum; DAT—direct agglutination test; *—band test intensity; μ—44 samples from VL/AIDS patients were not tested by DAT, Kalazar Detect and IT-Leish tests.
(DOCX)

**S4 Table. Collection date and diagnosis criteria for VL patients, VL /AIDS patients, and healthy controls from various Brazilian endemic areas (Panel 2).** n—number of samples; VL—visceral leishmaniasis; VL/AIDS—co-infection; DAT—direct agglutination test.
(DOCX)

**S5 Table. Collection date and diagnosis criteria for potentially cross-reactive controls.** n—number of samples.
(DOCX)

**S6 Table. Specificity of rK18-ELISA, rK28-ELISA, rK39-ELISA, and rKR95-ELISA in sera from patients with other diseases (Panel 3).** n—number of negative samples; 95% CI—95% probability confidence interval. Cochran's Q test: a—p < 0.0001; b—p = 0.0027; c—p = 0.0006. Pairwise McNemar's test: d—p = 0.0040—compared with rK28-ELISA and rKR95-ELISA; p = 0.0090—compared with rK39-ELISA. e—p = 0.0358—compared with rK39-ELISA and rKR95-ELISA; p = 0.0833—compared with rK28-ELISA. f—p = 0.0140—compared with rK28-ELISA and rKR95-ELISA. g—p = 0.0067—compared with rK28-ELISA, rK39-ELISA, and rKR95-ELISA. h—p < 0.0001—compared with rK28-ELISA, rK39-ELISA, and rKR95-ELISA. x—R cannot deal with factors containing only one level.
(DOCX)

**S1 Fig. Flow diagram of rK28-ELISA, rK39-ELISA, rK18-ELISA, and rKR95-ELISA results, obtained in serum samples from VL patients and healthy controls, employed to construct ROC (receiver operating characteristic) curves.** n—number of samples; VL—visceral leishmaniasis; DAT—direct agglutination test.
(TIF)

**S2 Fig. Flow diagram for reporting the results of rK28-ELISA, rK39-ELISA, rK18-ELISA, and rKR95-ELISA in serum samples from VL patients, VL/AIDS patients, and healthy controls from different geographical Brazilian regions.** n—number of samples; VL—visceral leishmaniasis; VL/AIDS—co-infection; DAT—direct agglutination test.
(TIF)

**S3 Fig. Flow diagram for reporting the results of rK28-ELISA, rK39-ELISA, rK18-ELISA, and rKR95-ELISA in serum samples from patients diagnosed with inflammatory disorders and other infectious diseases.** n—number of samples; VL—visceral leishmaniasis; VL/AIDS—co-infection; DAT—direct agglutination test.
(TIF)

**S4 Fig. Spearman correlation heat map.** The heat map shows the correlation between the results obtained by rK28-ELISA, rK39-ELISA, rK18-ELISA, and rKR95-ELISA, two-by-two, testing samples of VL patients (n = 90) and healthy endemic controls (n = 90) (Panel 1). The strength of the correlation is displayed in colors ranging from black to white.
(TIF)

**S5 Fig. Spearman correlation analysis heat map.** The heat map shows the correlation between the results obtained by rK18-ELISA, rK28-ELISA, rK39-ELISA, and rKR95-ELISA two-by-two. (A) One hundred twenty-two samples from VL patients, (B) 64 samples from VL / AIDS patients, (C) 83 samples from healthy controls, and (D) 190 samples from patients with

other infectious diseases. The strength of the correlation is displayed in colors ranging from black to white.
(TIF)

**S6 Fig. Sensitivity of rK28-ELISA, rK39-ELISA, rK18-ELISA, and rKR95-ELISA in samples from patients with visceral leishmaniasis from different geographical regions in Brazil (Panel 2).** *—p = 0.0074 (Cochran's Q test); p = 0.0455 (pairwise McNemar's test) compared with rK28-ELISA, rK39-ELISA, and rKR95-ELISA.
(TIF)

**S7 Fig. Sensitivity of rK28-ELISA, rK39-ELISA, rK18-ELISA, and rKR95-ELISA in samples from patients co-infected with visceral leishmaniasis and AIDS from different geographical regions in Brazil (Panel 2).**
(TIF)

**S8 Fig. Specificity of rK28-ELISA, rK39-ELISA, rK18-ELISA, and rKR95-ELISA in samples from healthy controls from different geographical regions in Brazil (Panel 2).** *—p < 0.0001 (Cochran's Q test); p = 0.0002 (pairwise McNemar's test) compared with rK28-E-LISA, rK39-ELISA, and rKR95-ELISA. **—p < 0.0001 (Cochran's Q test); p = 0.0016 (pairwise McNemar's test) compared with rK28-ELISA and rK39-ELISA; p = 0.0027 (pairwise McNemar's test) compared with rKR95-ELISA.
(TIF)

**S1 Dataset. Raw data obtained by ELISA with the recombinant antigens applied to the various groups of samples studied.** ND—not done; MI—missing information; M—male; F—female; Age—in years. DAT—direct agglutination test. ELISA binary result: 1—positive; 0—negative. Kalazar Detect and IT-Leish—rK39-based RDT: 0—negative; faint, 1, 2, 3—positive. Pos—positive; Neg—negative.
(XLSX)

**S1 Checklist. STARD checklist.**
(DOCX)

## Author Contributions

**Conceptualization:** Mahyumi Fujimori, Ruth Tamara Valencia-Portillo, Hiro Goto, Maria Carmen Arroyo Sanchez.

**Data curation:** Mahyumi Fujimori, Ruth Tamara Valencia-Portillo, Hiro Goto, Maria Carmen Arroyo Sanchez.

**Formal analysis:** Mahyumi Fujimori, Ruth Tamara Valencia-Portillo, Hiro Goto, Maria Carmen Arroyo Sanchez.

**Funding acquisition:** Hiro Goto, Maria Carmen Arroyo Sanchez.

**Investigation:** Mahyumi Fujimori, Ruth Tamara Valencia-Portillo, José Angelo Lauletta Lindoso, Beatriz Julieta Celeste, Roque Pacheco de Almeida, Carlos Henrique Nery Costa, Alda Maria da Cruz, Angelita Fernandes Druzian, Malcolm Scott Duthie, Carlos Magno Castelo Branco Fortaleza, Ana Lúcia Lyrio de Oliveira, Anamaria Mello Miranda Paniago, Igor Thiago Queiroz, Steve Reed, Aarthy C. Vallur, Hiro Goto, Maria Carmen Arroyo Sanchez.

**Methodology:** Mahyumi Fujimori, Ruth Tamara Valencia-Portillo, Hiro Goto, Maria Carmen Arroyo Sanchez.

**Project administration:** Hiro Goto, Maria Carmen Arroyo Sanchez.

**Resources:** Hiro Goto, Maria Carmen Arroyo Sanchez.

**Supervision:** Hiro Goto, Maria Carmen Arroyo Sanchez.

**Validation:** Mahyumi Fujimori, Ruth Tamara Valencia-Portillo, Hiro Goto, Maria Carmen Arroyo Sanchez.

**Visualization:** Mahyumi Fujimori, Ruth Tamara Valencia-Portillo, Hiro Goto, Maria Carmen Arroyo Sanchez.

**Writing – original draft:** Mahyumi Fujimori, Ruth Tamara Valencia-Portillo, Hiro Goto, Maria Carmen Arroyo Sanchez.

**Writing – review & editing:** Mahyumi Fujimori, Ruth Tamara Valencia-Portillo, Hiro Goto, Maria Carmen Arroyo Sanchez.

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
