## [Decision Letter · Decision Letter 0]

26 Dec 2022

PONE-D-22-28460Recombinant protein KR95 as an alternative for serological diagnosis of human visceral leishmaniasis in the AmericasPLOS ONE

Dear Dr. Sanchez,

Thank you for submitting your manuscript to PLOS ONE. After careful consideration, we feel that it has merit but does not fully meet PLOS ONE’s publication criteria as it currently stands. Therefore, we invite you to submit a revised version of the manuscript that addresses the points raised during the review process.

 This is an interesting and well-conducted study, with an adequate number of figures and supplemental materials. However, in light of the reviewers' comments, it is necessary to further discuss the limitations of the research, especially the use of monoplicates and the non-adoption of PCR in negative samples.I also suggest that the "diagnostic odds ratios" (DORs) of the panel 1 data and the confidence intervals of the values presented in the abstract be included in the article.

We look forward to receiving your revised manuscript.

Kind regards,

Vinícius Silva Belo

Academic Editor

PLOS ONE

Journal Requirements:

“This study was supported by Laboratório de Investigação Médica (LIM-38) Hospital das Clínicas da Faculdade de Medicina da Universidade de Sao Paulo (https://limhc.fm.usp.br/portal/). HG received a research fellowship from Conselho Nacional de Desenvolvimento Científico e Tecnológico – CNPq (https://www.gov.br/cnpq/pt-br). MF received a scholarship (n°: 2017/03367-3) from Fundação de Amparo à Pesquisa do Estado de São Paulo (https://fapesp.br/). RTVP received a scholarship (n°: 88882.376665/2019-01 and n°: 88887.689454/2022-00) from Coordenação de Aperfeiçoamento de Pessoal de Nível Superior (https://www.gov.br/capes/pt-br). The funders had no role in study design, data collection and analysis, decision to publish, or preparation of the manuscript.”

“This study was supported by Laboratório de Investigação Médica (LIM-38) Hospital das Clínicas da Faculdade de Medicina da Universidade de Sao Paulo (https://limhc.fm.usp.br/portal/). HG received a research fellowship from Conselho Nacional de Desenvolvimento Científico e Tecnológico – CNPq (https://www.gov.br/cnpq/pt-br). MF received a scholarship (n°: 2017/03367-3) from Fundação de Amparo à Pesquisa do Estado de São Paulo (https://fapesp.br/). RTVP received a scholarship (n°: 88882.376665/2019-01 and n°: 88887.689454/2022-00) from Coordenação de Aperfeiçoamento de Pessoal de Nível Superior (https://www.gov.br/capes/pt-br). The funders had no role in study design, data collection and analysis, decision to publish, or preparation of the manuscript.”

5. We note that Figure 1 in your submission contain map images which may be copyrighted. All PLOS content is published under the Creative Commons Attribution License (CC BY 4.0), which means that the manuscript, images, and Supporting Information files will be freely available online, and any third party is permitted to access, download, copy, distribute, and use these materials in any way, even commercially, with proper attribution. For these reasons, we cannot publish previously copyrighted maps or satellite images created using proprietary data, such as Google software (Google Maps, Street View, and Earth). For more information, see our copyright guidelines: http://journals.plos.org/plosone/s/licenses-and-copyright.

a. You may seek permission from the original copyright holder of Figure(s) [#] to publish the content specifically under the CC BY 4.0 license. 

Reviewers' comments:

Reviewer's Responses to Questions

**Comments to the Author**

1. Is the manuscript technically sound, and do the data support the conclusions?

Reviewer #1: Yes

Reviewer #2: Yes

Reviewer #3: Partly

2. Has the statistical analysis been performed appropriately and rigorously? 

Reviewer #1: Yes

Reviewer #2: Yes

Reviewer #3: Yes

3. Have the authors made all data underlying the findings in their manuscript fully available?

Reviewer #1: Yes

Reviewer #2: Yes

Reviewer #3: Yes

4. Is the manuscript presented in an intelligible fashion and written in standard English?

Reviewer #1: Yes

Reviewer #2: Yes

Reviewer #3: Yes

5. Review Comments to the Author

Reviewer #1: The study was well conducted and the data support the results presented. The data is robust, clear and well presented graphically. Supplementary material has been presented and allows you to resolve any doubts you may have.

Reviewer #2: PONE-D-22-28460

The paper describes using two new recombinant antigens K18 and KR95 on VL-ELISA diagnosis. The manuscript is well present and can contribute to the use of the antigens in new variations of immunodiagnostic assays for human visceral leishmaniasis. I recommend the publication after minor revision.

M&M

- please clarify the DAT test used as a standard test (produced by? Which antigen is used in this assay? How about the sensibility and specificity of this assay?).

- Line 165- Why the negative samples were not confirmed by molecular assays like PCR or qPCR? The use of molecular assay is important because the health donors are from endemic regions.

- Line 200- Recombinant antigens-ELISA-Please describe the molecular differences between the recombinant antigens. Are they derived from the same original protein or are each of one are from different background proteins?

Discussion

Lines 624-628 – the authors did not add discussion in this paragraph. I don-t know if the following paragraph is about it. If it is, please unify both paragraphs.

Reviewer #3: The manuscript presents highly relevant public health findings, is very well drafted and meets the standards of the journal. I have a couple of questions:

1. Were the standardization, assessment and validation tests carried out on monoplicate? Were any reproducibility and repeatability tests conducted?

2. Was there a sample blinding plan for carrying out the tests?

3. For the Panel 2: serological tests associated with a compatible clinical are sufficient for the treatment of VL cases in Brazil.However, for use in serological panels, it is important to track these cases and obtain the final diagnostic outcome of these patients, whether due to therapeutic success or notification confirmation. Has this information been verified? If so, I think it's interesting to include.

4. There is significant heterogeneity in Panel 2. I believe that to obtain relevant epidemiological information, a more uniform and representative number of samples from each population would be needed. The tests used to characterize the panels should also be standardized. For VL/AIDS co-infected patients were characterized by the routine diagnostic methods and this could lead to misinterpretation. In this way, I suggest future tests, more homogeneous for validation with this objective. Would it fit as a limitation of the study?

5. Regarding the text, I only noticed a writing error: line 344: FN-false positive

6. I suggest reducing the number of figures.

6. PLOS authors have the option to publish the peer review history of their article (what does this mean?). If published, this will include your full peer review and any attached files.

Reviewer #1: No

Reviewer #2: **Yes: **Juliana Lopes Rangel Fietto

Reviewer #3: No

---

## [Author Response · Author response to Decision Letter 0]

2 Feb 2023

Dear editor and reviewers,

Thank you for carefully analyzing our manuscript and for your comments and suggestions. In the following paragraphs, we will reply to your answers and suggestions.

Academic editor comments: 

1. Discuss the limitations of the research, especially the use of monoplicates and the non-adoption of PCR in negative samples.

Answer: Regarding the use of monoplicates in the study, we wrote: "The sample dilutions in duplicate (50 µL/well) were applied…" but for clarity, we also added: "Throughout the study, samples were assayed in duplicate."

Regarding the non-adoption of PCR in negative samples, we considered three points: 

-Different proposed targets for Leishmania DNA detection exist, with different results (Galluzzi et al., 2018, DOI: 10.1186/s13071-018-2859-8). 

- The low number of circulating parasites in asymptomatic infections may influence the reproducibility of the molecular test (Santos Marques et al., 2012, DOI: 10.1371/journal.pntd.0001955).

-The serology may perform similarly to PCR (Lopes et al., 2023, DOI: 10.1007/s00436-022-07770-7).

This way, we did not consider that screening healthy control samples with DAT is a limitation of the research.

2. I also suggest that the "diagnostic odds ratios" (DORs) of the panel 1 data and the confidence intervals of the values presented in the abstract be included in the article.

Answer: We accepted your suggestion, included the confidence intervals of the values presented in the abstract, and calculated the "diagnostic odds ratios" of panels 1 and 2. 

3. Answer: We accepted your suggestion.

Answer: All figures are according to PACE.

5. If applicable, we recommend that you deposit your laboratory protocols in protocols.io to enhance the reproducibility of your results. 

Answer: Thank you for the suggestion.

Journal Requirements:

Answer: We followed the style requirements and made some changes.

2. Please provide additional details regarding participant consent.

Answer: We followed your suggestion and wrote: "Since the 639 samples employed in this study were obtained from the Biorepository belonging to the Instituto de Medicina Tropical, Faculdade de Medicina, USP, and under the responsibility of Hiro Goto and Maria Carmen Arroyo Sanchez, the need for participant consent was waived by the ethics committee. The samples were coded to preserve the individual's anonymity. Furthermore, the research results were disclosed so that the individual could not be correlated with the said result. We also guarantee the ethically correct use of the material and the information obtained from it."

3. We note that the grant information you provided in the 'Funding Information' and 'Financial Disclosure' sections do not match.

When you resubmit, please ensure that you provide the correct grant numbers for the awards you received for your study in the 'Funding Information' section.

Answer: We accepted your suggestion.

Please remove any funding-related text from the manuscript and let us know how you would like to update your Funding Statement. 

Answer: We removed the funding information from the Acknowledgements Section of the manuscript and will add it to the cover letter.

5. We note that Figure 1 in your submission contain map images which may be copyrighted. 

Answer: In Figure 1, the map was replaced by another obtained from https://apps.nationalmap.gov/viewer/.

6. Please review your reference list to ensure that it is complete and correct. 

Answer: References were checked and reviewed.

Reviewers' comments:

Reviewer #1: The study was well conducted and the data support the results presented. The data is robust, clear and well-presented graphically. Supplementary material has been presented and allows you to resolve any doubts you may have.

Answer: Thank you for carefully analyzing our manuscript and for your comments.

Reviewer #2: The paper describes using two new recombinant antigens K18 and KR95 on VL-ELISA diagnosis. The manuscript is well present and can contribute to the use of the antigens in new variations of immunodiagnostic assays for human visceral leishmaniasis. I recommend the publication after minor revision.

Answer: Thank you for carefully analyzing our manuscript and for your comments and suggestions.

M&M

1. please clarify the DAT test used as a standard test (produced by? Which antigen is used in this assay? How about the sensibility and specificity of this assay?).

Answer: The use of DAT (KIT Biomedical Research in Amsterdam, The Netherlands) as a reference test for assembling the health endemic control panels was supported by the good sensitivity (CI 95%)/specificity (CI 95%) reported in the literature. For instance, 98.2% (90.4-99.9)/100.0% (92.6-100.0) [18]; 96.6% (89.7-99.1)/98.1% (92.6-99.7) [27] and 88.5% (84.1-92.0)/95.4 (89.2-98.1) [26]. We also included in the manuscript the following:

" Direct Agglutination Test (DAT) based on L. donovani promastigotes was produced by KIT Biomedical Research – Amsterdam, The Netherlands.” 

“For DAT, samples were two-fold diluted from 1:100 (final 1:200) through 1:102,400 (final 1:204,800), and a cut-off of 1:3,200 was adopted [18,23,26,27]. All healthy endemic control samples were DAT negative (titer < 1:3,200). However, VL patients with negative DAT were included in the study as long as they were positive in the parasitological test.”

2. Line 165- Why the negative samples were not confirmed by molecular assays like PCR or qPCR? The use of molecular assay is important because the health donors are from endemic regions.

Answer: Regarding using molecular assays to detect Leishmania DNA, we did not perform them on positive or negative control samples from the endemic area. In this study, we used only samples from our biorepository, and we did not have plasma or buffy coat to carry out molecular assays. PCR is always considered very sensitive, but we need a standardized and validated robust method for molecular diagnosis. Searching in the specific literature, different proposed targets for Leishmania DNA detection exist, with different results, as seen in the review article by Galluzzi et al. (2018) (DOI: 10.1186/s13071-018-2859-8). As in asymptomatic infections, the low number of circulating parasites may influence the reproducibility of the molecular test (Santos Marques et al., 2012, DOI: 10.1371/journal.pntd.0001955) and considering that the serology may perform similarly to PCR (Lopes et al., 2023, DOI: 10.1007/s00436-022-07770-7) we did not see that the screening of healthy control samples with DAT is a limitation of the study. 

3. Line 200- Recombinant antigens-ELISA-Please describe the molecular differences between the recombinant antigens. Are they derived from the same original protein or are each of one are from different background proteins?

Answer: We accepted your suggestion and included information about the recombinant antigens: 

"rK39 (L. infantum – syn. chagasi) is part of a large protein kinesin-related (Lc-Kin), containing 298 amino acids and has a molecular mass of 38.9 kD [15]. rK28 (L. donovani) is a fusion polyprotein comprising HASPB1 (L. infantum K26 homolog), LdK39 (L. infantum K39 homolog), and HASPB2 (L. infantum K9 homolog) and has a molecular mass of 28.33 [19]. rKR95 (L. donovani) is a kinesin-related protein with a molecular mass of 95 kD, presenting 100% identity with L. infantum [28]. rK18 (L. infantum – syn. chagasi) is a tandem repeat hypothetical protein (also known as rTR18) with a molecular mass of 18 kD, presenting 100% identity with L. donovani [29]."

Discussion

4. Lines 624-628 – the authors did not add discussion in this paragraph. I don't know if the following paragraph is about it. If it is, please unify both paragraphs.

Answer: We accepted your suggestion and unified both paragraphs.

Reviewer #3: The manuscript presents highly relevant public health findings, is very well drafted and meets the standards of the journal. I have a couple of questions:

Answer: Thank you for carefully analyzing our manuscript and for your comments and suggestions.

1. Were the standardization, assessment and validation tests carried out on monoplicate? Were any reproducibility and repeatability tests conducted?

Answer: The tests were assayed in duplicates. We added: "Throughout the study, samples were assayed in duplicate."

We did not conduct tests on reproducibility and repeatability in this study. However, Fujimori calculated the coefficient of variation of the repeatability and reproducibility using the same antigens in dog samples, as described below: 

For repeatability, 30 replicates of a positive sample for L. infantum infection were assayed in the same plate: rK28 = 3.8%, rK39 = 4.4%, rK18 = 5.0%, and rKR95 = 4.7%. For reproducibility, 10 replicates of a positive sample for L. infantum infection were tested in five subsequent days: rK28 = 13.4%, rK39 = 10.5%, rK18 = 17.1%, and rKR95 = 10.9%. (Fujimori, M. (2019). Validação de ensaio imunoenzimático (ELISA) com antígenos recombinantes no diagnóstico sorológico da infecção canina por Leishmania infantum (Doctoral dissertation, Universidade de São Paulo).

2. Was there a sample blinding plan for carrying out the tests?

Answer: Although the samples were not blinded, they were tested randomly and diluted in mini tubes using multichannel pipets to apply the diluted samples. "The samples were coded and analyzed randomly to prevent the risk of bias." 

3. For the Panel 2: serological tests associated with a compatible clinical are sufficient for the treatment of VL cases in Brazil. However, for use in serological panels, it is important to track these cases and obtain the final diagnostic outcome of these patients, whether due to therapeutic success or notification confirmation. Has this information been verified? If so, I think it's interesting to include.

Answer: Considering VL patients from panel 2, we added the following information missing in the manuscript: " The active leishmaniasis was confirmed at the collection site before they were included in the study, and all patients presented at least two of the following symptoms or signs of VL: fever, hepatomegaly, splenomegaly, and cytopenia." 

4. There is significant heterogeneity in Panel 2. I believe that to obtain relevant epidemiological information, a more uniform and representative number of samples from each population would be needed. The tests used to characterize the panels should also be standardized. For VL/AIDS co-infected patients were characterized by the routine diagnostic methods and this could lead to misinterpretation. In this way, I suggest future tests, more homogeneous for validation with this objective. Would it fit as a limitation of the study?

Answer: We agreed with your comment and included it as a limitation of the study. We wrote: " Another limitation was the heterogeneity of the groups of samples in panel 2, with a smaller number of healthy controls in Aracaju and Bauru, and the characterization of VL/AIDS co-infected patients that were different according to the locality of collection." 

5. Regarding the text, I only noticed a writing error: line 344: FN-false positive

Answer: Thank you for the correction. 

6. I suggest reducing the number of figures.

Answer: We consider all figures important for the clarity of our entire data.

---

## [Decision Letter · Decision Letter 1]

16 Feb 2023

Recombinant protein KR95 as an alternative for serological diagnosis of human visceral leishmaniasis in the Americas

PONE-D-22-28460R1

Dear Dr. Sanchez,

We’re pleased to inform you that your manuscript has been judged scientifically suitable for publication and will be formally accepted for publication once it meets all outstanding technical requirements.

Kind regards,

Vinícius Silva Belo

Academic Editor

PLOS ONE

Additional Editor Comments (optional):

Reviewers' comments:

Reviewer's Responses to Questions

**Comments to the Author**

1. If the authors have adequately addressed your comments raised in a previous round of review and you feel that this manuscript is now acceptable for publication, you may indicate that here to bypass the “Comments to the Author” section, enter your conflict of interest statement in the “Confidential to Editor” section, and submit your "Accept" recommendation.

Reviewer #3: All comments have been addressed

2. Is the manuscript technically sound, and do the data support the conclusions?

Reviewer #3: Yes

3. Has the statistical analysis been performed appropriately and rigorously? 

Reviewer #3: Yes

4. Have the authors made all data underlying the findings in their manuscript fully available?

Reviewer #3: Yes

5. Is the manuscript presented in an intelligible fashion and written in standard English?

Reviewer #3: Yes

6. Review Comments to the Author

Reviewer #3: (No Response)

7. PLOS authors have the option to publish the peer review history of their article (what does this mean?). If published, this will include your full peer review and any attached files.

Reviewer #3: **Yes: **Marcelino, AP

---

## [Editor Report · Acceptance letter]

22 Feb 2023

PONE-D-22-28460R1 

Recombinant protein KR95 as an alternative for serological diagnosis of human visceral leishmaniasis in the Americas 

Dear Dr. Sanchez:

I'm pleased to inform you that your manuscript has been deemed suitable for publication in PLOS ONE. Congratulations! Your manuscript is now with our production department. 

Kind regards, 

on behalf of

Dr. Vinícius Silva Belo 

Academic Editor

PLOS ONE